# Large-Scale Price Optimization via Network Flow

**Shinji Ito**
NEC Corporation
s-ito@me.jp.nec.com

**Ryohei Fujimaki**
NEC Corporation
rfujimaki@nec-labs.com

## Abstract

This paper deals with price optimization, which is to find the best pricing strategy that maximizes revenue or profit, on the basis of demand forecasting models. Though recent advances in regression technologies have made it possible to reveal price-demand relationship of a large number of products, most existing price optimization methods, such as mixed integer programming formulation, cannot handle tens or hundreds of products because of their high computational costs. To cope with this problem, this paper proposes a novel approach based on network flow algorithms. We reveal a connection between supermodularity of the revenue and cross elasticity of demand. On the basis of this connection, we propose an efficient algorithm that employs network flow algorithms. The proposed algorithm can handle hundreds or thousands of products, and returns an exact optimal solution under an assumption regarding cross elasticity of demand. Even if the assumption does not hold, the proposed algorithm can efficiently find approximate solutions as good as other state-of-the-art methods, as empirical results show.

## 1 Introduction

Price optimization is a central research topic with respect to revenue management in marketing science [10, 16, 18]. The goal is to find the best price strategy (a set of prices for multiple products) that maximizes revenue or profit. There is a lot of literature regarding price optimization [1, 5, 10, 13, 17, 18, 20], and significant success has been achieved in industries such as online retail [7], fast-fashion [5], hotels [13, 14], and airlines [16]. One key component in price optimization is demand modeling, which reveals relationships between price and demand. Though traditional studies have focused more on a single price-demand relationship, such as price elasticity of demand [13, 14] and the law of diminishing marginal utility [16], multi-product relationships such as cross price elasticity of demand [15] have recently received increased attention [5, 17]. Recent advances in regression technologies (non-linear, sparse, etc.) make demand modeling over tens or even hundreds of products possible, and data oriented demand modeling has become more and more important.

Given demand models of multiple products, the role of optimization is to find the best price strategy. Most existing studies for multi-product price optimization employ mixed-integer programming [5, 13, 14] due to the discrete nature of individual prices, but their methods cannot be applied to large scale problems with tens or hundreds of products since their computational costs exponentially increases over increasing numbers of products. Though restricting demand models might make optimization problems tractable [5, 7], such approaches cannot capture complicated price-demand relationships and often result in poor performance. Ito and Fujimaki [9] have recently proposed a *prescriptive price optimization* framework to efficiently solve multi-product price optimization with non-linear demand models. In this prescriptive price optimization, the problem is transformed into a sort of binary quadratic programming problem, and they have proposed an efficient relaxation method based on semi-definite programming (SDP). Although their approach has significantly improved computational efficiency over that of mixed-integer approaches, the computational complexity of their SDP formulation requires $O(M^6)$ in theory, where $M$ is the number of products, and it is not

sufficiently scalable for large scale problems with hundreds of products, as our empirical evaluation show in Section 5.

The goal of this paper is to develop an efficient algorithm for large scale multi-product price optimization problems that can handle hundreds of products as well as flexible demand models. Our main technical contributions are two-fold. First, we reveal the connection between submodularity of the revenue and cross elasticity of demand. More specifically, we show that the gross profit function of the prescriptive price optimization is supermodular (i.e., the maximization of the gross profit function is equivalent to the submodular minimization) under the assumption regarding cross elasticity of demand that there are no pairs of complementary goods (we refer to this property as a *substitute-goods property*).[1] On the basis of the submodularity, we propose a practical, efficient algorithm that employs network flow algorithms for minimum cut problems and returns exact solutions for problems with the substitute-goods property. Further, even in cases in which the property does not hold, it can efficiently find approximate solutions by iteratively improving submodular lower bounds. Our empirical results show that the proposed algorithm can successfully handle hundreds of products and derive solutions as good as other state-of-the-art methods, while its computational cost is much cheaper, regardless of whether the substitute-goods property holds or not.

## 2    Literature review

Our price optimization problems are reduced to binary quadratic problems such as (4). It is well known that submodular binary quadratic programming problems can be reduced to minimum cut problems [12], and hence it can be solved by maximum flow algorithms. Also for unconstrained non-submodular binary quadratic programming problems, there is a lot of literature regarding optimization algorithm using minimum cut, especially in the context of Markov random fields inference or energy minimization in computer vision [2, 3, 4, 8, 11, 22]. Above all, QPBO method [2, 11] and its extensions such as QPBOI method [19] are known to be state-of-the-art methods in terms of scalability and theoretical properties. These QPBO/QPBOI and our method are similar in that they all employ network flow algorithms and derive not only partial/approximate solutions but also lower bounds of the exact optimal (minimum) value. Our methods, however, differs from QPBO and its extensions in network structures, accuracy and scalability, as is shown in Section 5.

## 3    Price optimization and submodularity in cross elasticity of demand

Suppose we have $M$ products and a product index is denoted by $i \in \{1, \ldots, M\}$. In prescriptive price optimization [9], for a price strategy $\mathbf{p} = [p_1, \ldots, p_M]^\top$, where $p_i$ is the price of the $i$-th product, and for external variables $\mathbf{r} = [r_1, \ldots, r_D]^\top$ such as weather, temperature and days of the week, the sales quantity (demand) for the $i$-th product is modeled by the following regression formula:

$$q_i(\mathbf{p}, \mathbf{r}) = \sum_{j=1}^{M} f_{ij}(p_j) + \sum_{t=1}^{D} g_{it}(r_t), \tag{1}$$

where $f_{ii}$ expresses the effect of price elasticity of demand, $f_{ij}$ $(i \neq j)$ reflects the effect of cross elasticity, and $g_{it}$ represent how the $t$-th external variable affect the sales quantity. Note that $f_{ij}$ for all $(i, j)$ can be arbitrary functions, and Eq. (1) covers various regression (demand) models, such as linear regression, additive models [21], linear regression models with univariate basis functions, etc. This paper assumes that the regression models are given using existing methods and focuses its discussion on optimization.

Given $q_i(\mathbf{p})$ for all i and a cost vector $\mathbf{c} = [c_1, \ldots, c_M]^\top$, and fixed external variables $\mathbf{r}$, the *gross profit* can be represented as

$$\ell(\mathbf{p}) = \sum_{i=1}^{M} (p_i - c_i) q_i(\mathbf{p}) = \sum_{i=1}^{M} (p_i - c_i) \left( \sum_{j=1}^{M} f_{ij}(p_j) + \sum_{t=1}^{D} g_{it}(r_t) \right). \tag{2}$$

The goal of price optimization is to find $\mathbf{p}$ maximizing $\ell(\mathbf{p})$. In practice, $p_m$ is often chosen from the finite set $\mathcal{P}_i = \{P_{i1}, \ldots, P_{iK}\} \subseteq \mathbb{R}$ of $K$ price candidates, where $P_{iK}$ might be a list price and $P_{ik}$ $(k < K)$ might be discounted prices such as 10%-off, 5%-off, 3%-off. Then, the problem of maximizing the gross profit can be formulated as the following combinatorial optimization problem:

$$\text{Maximize} \quad \ell(\mathbf{p}) \quad \text{subject to} \quad p_i \in \mathcal{P}_i. \tag{3}$$

It is trivial to show that (3) is NP-hard in general.

Let us formally define the "substitute-goods property" as follows.

**Definition 1** (Substitute-Goods Property). The demand model defined by (1) of the $i$-th product is said to satisfy the *substitute-goods property* if $f_{ij}$ is monotone non-decreasing for all $j \neq i$.

The concept of substitute-goods property is practical and important because retailers often deal with substitute goods. Suppose the situation that a retailer decides a price strategy of different brand in the same products category. For example, supermarkets sell milk of different brands and car dealerships sell various types of cars. These products are usually substitute goods. This kind of cross elasticity effect is one of advanced topics in revenue management and is practically important [13, 14, 17]. Our key observation is the connection between the substitute-goods property in marketing science and the supermodularity of the gross profit function, which is formally described in the following proposition.

**Proposition 2.** *The gross profit function $\ell : \mathcal{P}_1 \times \cdots \times \mathcal{P}_M \to \mathbb{R}$ is supermodular[2] if demand models defined by (1) for all products satisfies the substitute-goods property.*

The above proposition implies that, under the assumption of the substitute-goods property, problem (3) can be solved precisely using submodular minimization algorithms, where time complexity is a polynomial in $M$ and $K$. This fact, however, does not necessarily imply that there exists a practical, efficient algorithm for problem (3). Indeed, general submodular minimization algorithms are slow in practice even though their time complexities are polynomial. Further, actual models do not always satisfy the substitute-goods property. We propose solutions to these problems in the next section.

## 4 Network flow-based algorithm for revenue maximization

### 4.1 Binary quadratic programming formulation

This section shows that problem (3) can be reduced to the following binary quadratic programming problem (notations are explained in the latter part of this section):

$$\begin{aligned}
\text{Minimize} \quad & \mathbf{x}^\top A \mathbf{x} + \mathbf{b}^\top \mathbf{x} \\
\text{subject to} \quad & \mathbf{x} = [x_1, \ldots, x_n]^\top \in \{0, 1\}^n, \\
& x_u \leq x_v \quad ((u, v) \in C),
\end{aligned} \tag{4}$$

Each variable $p_i$ takes $P_{ik}$ if and only if the binary vector $\mathbf{x}_i = [x_{i1}, \ldots, x_{i,K-1}]^\top \in \{0, 1\}^{(K-1)}$ satisfies:

$$\mathbf{x}_i = \mathbf{c}_k := [\underbrace{1, \ldots, 1}_{k-1}, \underbrace{0, \ldots, 0}_{K-k}]^\top \quad (k = 1, \ldots, K). \tag{5}$$

Also we define $\mathbf{x} = [\mathbf{x}_1^\top, \ldots, \mathbf{x}_M^\top]^\top \in \{0, 1\}^{(K-1)M}$ and redefine the indices of the entries of $\mathbf{x}$ as $\mathbf{x} = [x_1, x_2, \ldots, x_{(K-1)M}]$, i.e. $x_{i,k} = x_{i(K-1)+k}$ for notational simplicity.

Defining $\ell_{ij} : \mathcal{P}_i \times \mathcal{P}_j \to \mathbb{R}$ by $\ell_{ij}(p_i, p_j) = (p_i - c_i)f_{ij}(p_j)$ for $i \neq j$ and $\ell_i : \mathcal{P}_i \to \mathbb{R}$ by $\ell_i(p_i) = (p_i - c_i)(f_{ii}(p_i) + \sum_{t=1}^D g_{it}(r_t))$, we can express $\ell$ as

$$\ell(\mathbf{p}) = \sum_{1 \leq i, j \leq M, i \neq j} \ell_{ij}(p_i, p_j) + \sum_{i=1}^M \ell_i(p_i). \tag{6}$$

**Algorithm 1** s-t cut for price optimization with the substitute-goods property

---
**Input:** Problem instance $(A, \mathbf{b}, C)$ of (4), where all entries of $A$ are non-positive.
**Output:** An optimal solution $\mathbf{x}^*$ to (4).
  1: Construct a weighted directed graph $G = (V, E, w)$ satisfying (9).
  2: Add edges $C$ with weight $\infty$ to $G$, i.e., set $E \leftarrow E \cup C$ and $w(u, v) \leftarrow \infty$ for all $(u, v) \in C$.
  3: Compute a minimum $s$-$t$ cut $U^*$ of $G$, define $\mathbf{x}^*$ by (10) and return $\mathbf{x}^*$.

---

Using $\mathbf{x}_i$, we can construct matrices $A_{ij} \in \mathbb{R}^{(K-1)\times(K-1)}$ for which it holds that

$$\ell_{ij}(p_i, p_j) = -\mathbf{x}_i^\top A_{ij}\mathbf{x}_j + \text{const.} \tag{7}$$

Indeed, matrices $A_{ij} = [a_{uv}^{ij}]_{1 \le u,v \le K-1} \in \mathbb{R}^{(K-1)\times(K-1)}$ defined by

$$a_{uv}^{ij} = -\ell_{ij}(P_{i,u+1}, P_{j,v+1}) + \ell_{ij}(P_{i,u}, P_{j,v+1}) + \ell_{ij}(P_{i,u+1}, P_{j,v}) - \ell_{ij}(P_{i,u}, P_{j,v}) \tag{8}$$

satisfy (7). In a similar way, we can construct $\mathbf{b}_i \in \mathbb{R}^{K-1}$ such that $\ell_i(p_i) = -\mathbf{b}_i^\top \mathbf{x}_i + \text{const.}$ Accordingly, the objective function $\ell$ of problem (3) satisfies $\ell(\mathbf{p}) = -(\mathbf{x}^\top A\mathbf{x} + \mathbf{b}^\top \mathbf{x}) + \text{const.}$, where we define $A = [A_{ij}]_{1 \le i,j \le M} \in \mathbb{R}^{(K-1)M\times(K-1)M}$ and $\mathbf{b} = [\mathbf{b}_i]_{1 \le i \le M} \in \mathbb{R}^{(K-1)M}$. The conditions $\mathbf{x}_i \in \{\mathbf{c}_1, \ldots, \mathbf{c}_K\}$ $(i = 1, \ldots, M)$ can be expressed as $x_u \le x_v$ $((u, v) \in C)$, where we define $C := \{((K-1)(i-1) + k + 1, (K-1)(i-1) + k) \mid 1 \le i \le M, 1 \le k \le K-2\}$. Consequently, problem (3) is reduced to problem (4). Although [9] also gives another BQP formulation for the problem (3) and relaxes it to a semi-definite programming problem, our construction of the BQP problem can be solved much more efficiently, as is explained in the next section.

## 4.2 Minimum cut for problems with substitute goods property

As is easily seen from (8), if the problem satisfies the substitute-goods property, matrix $A$ has only non-positive entries. It is well known that unconstrained binary quadratic programming problems such as (4) with non-positive $A \in \mathbb{R}^{n \times n}$ and $C = \emptyset$ can be efficiently solved[3] by algorithms for minimum cut [6]. Indeed, we can construct a positive weighted directed graph, $G = (V = \{s, t, 1, 2, \ldots, n\}, E \subseteq V \times V, w : E \to \mathbb{R}_{>0} \cup \{\infty\})$[4] for which

$$\mathbf{x}^\top A\mathbf{x} + \mathbf{b}^\top \mathbf{x} = c_G(\{s\} \cup \{u \mid x_u = 1\}) + \text{const} \tag{9}$$

holds for all $\mathbf{x} \in \{0, 1\}^n$, where $c_G$ is the cut function of graph $G$[5]. Hence, once we can compute a minimum $s$-$t$ cut $U$ that is a vertex set $U \subseteq V$ minimizing $c_G(U)$ subject to $s \in U$ and $t \notin U$, we can construct an optimal solution $\mathbf{x} = [x_1, \ldots, x_n]^\top$ to the problem (4) by setting

$$x_u = \begin{cases} 1 & (u \in U) \\ 0 & (u \notin U) \end{cases} \quad (u = 1, \ldots, n). \tag{10}$$

For constrained problems such as (4) with $C \ne \emptyset$, the constraint $x_u \le x_v$ is equivalent to $x_u = 1 \implies x_v = 1$. This condition can be, in the minimum cut problem, expressed as $u \in U \implies v \in U$. By adding a directed edge $(u, v)$ with weight $\infty$, we can forbid the minimum cut to violate the constraints. In fact, if both $u \in U$ and $v \notin U$ hold, the value of the cut function is $\infty$, and hence such a $U$ cannot be a minimum cut. We summarize this in Algorithm 1.

## 4.3 Submodular relaxation for problems without the substitute-goods property

For problems without the substitute-goods property, we first decompose the matrix $A$ into $A^+$ and $A^-$ so that $A^+ + A^- = A$, where $A^+ = [a_{uv}^+]$ and $A^- = [a_{uv}^-] \in \mathbb{R}^{n \times n}$ are given by

$$a_{uv}^+ = \begin{cases} a_{uv} & (a_{uv} \ge 0) \\ 0 & (a_{uv} < 0) \end{cases}, \quad a_{uv}^- = \begin{cases} 0 & (a_{uv} \ge 0) \\ a_{uv} & (a_{uv} < 0) \end{cases} \quad (u, v \in N). \tag{11}$$

This leads to a decomposition of the objective function of Problem (4) into supermodular and submodular terms:

$$\mathbf{x}^\top A \mathbf{x} + \mathbf{b}^\top \mathbf{x} = \mathbf{x}^\top A^+ \mathbf{x} + \mathbf{x}^\top A^- \mathbf{x} + \mathbf{b}^\top \mathbf{x}, \tag{12}$$

where $\mathbf{x}^\top A^+ \mathbf{x}$ is supermodular and $\mathbf{x}^\top A^- \mathbf{x} + \mathbf{b}^\top \mathbf{x}$ is submodular. Our approach is to replace the supermodular term $\mathbf{x}^\top A^+ \mathbf{x}$ by a linear function to construct a submodular function approximating $\mathbf{x}^\top A \mathbf{x} + \mathbf{b}^\top \mathbf{x}$, that can be minimized by Algorithm 1. Similar approaches can be found in the literature, e.g. [8, 22], but ours has a significant point of difference; our method constructs approximate functions bounding objectives *from below*, which provides information about the degree of accuracy.

Consider an affine function $h(\mathbf{x})$ such that $h(\mathbf{x}) \leq \mathbf{x}^\top A^+ \mathbf{x}$ for all $\mathbf{x} \in \{0,1\}^n$. Such an $h$ can be constructed as follows. Since

$$\gamma_{uv}(x_u + x_v - 1) \leq x_u x_v \quad (x_u, x_v \in \{0,1\}) \tag{13}$$

holds for all $\gamma_{uv} \in [0,1]$, an arbitrary matrix $\Gamma \in [0,1]^{n \times n}$ satisfies

$$\mathbf{x}^\top A^+ \mathbf{x} \geq \mathbf{x}^\top (A^+ \circ \Gamma)\mathbf{1} + \mathbf{1}^\top (A^+ \circ \Gamma)\mathbf{x} - \mathbf{1}^\top (A^+ \circ \Gamma)\mathbf{1} =: h_\Gamma(\mathbf{x}), \tag{14}$$

where $A^+ \circ \Gamma$ denotes the Hadamard product, i.e., $(A^+ \circ \Gamma)_{uv} = a_{uv}^+ \cdot \gamma_{uv}$. From inequality (14), the optimal value of the following problem,

$$\begin{array}{ll} \text{Minimize} & \mathbf{x}^\top A^- \mathbf{x} + \mathbf{b}^\top \mathbf{x} + h_\Gamma(\mathbf{x}) \\ \text{subject to} & \mathbf{x} = [x_1, \ldots, x_n]^\top \in \{0,1\}^n, \\ & x_u \leq x_v \quad ((u,v) \in C), \end{array} \tag{15}$$

is a lower bound for that of problem (4). Since $A^-$ has non-positive entries and $\mathbf{b}^\top \mathbf{x} + h_\Gamma(\mathbf{x})$ is affine, we can solve (15) using Algorithm 1 to obtain an approximate solution for (4) and a lower bound for the optimal value of (4).

## 4.4 Proximal gradient method with sequential submodular relaxation

An essential problem in submodular relaxation is how to choose $\Gamma \in [0,1]^{n \times n}$ and to optimize $\mathbf{x}$ given $\Gamma$. Let $\psi(\Gamma)$ denote the optimal value of (15), i.e., define $\psi(\Gamma)$ by $\psi(\Gamma) = \min_{\mathbf{x} \in R} \mathbf{x}^\top A^- \mathbf{x} + \mathbf{b}^\top \mathbf{x} + h_\Gamma(\mathbf{x})$, where $R$ is the feasible region of (15). Then, for simultaneous optimization of $\mathbf{x}$ and $\Gamma$, we consider the following problem:

$$\text{Maximize } \psi(\Gamma) \quad \text{subject to } \Gamma \in [0,1]^{n \times n}, \tag{16}$$

which can be rewritten as follows:[6]

$$\text{Minimize } -\psi(\Gamma) + \Omega(\Gamma) \quad \text{subject to } \Gamma \in \mathbb{R}^{n \times n}, \tag{17}$$

where we define $\Omega : \mathbb{R}^{n \times n} \to \mathbb{R} \cup \{\infty\}$ by

$$\Omega(\Gamma) = \begin{cases} 0 & (\Gamma \in [0,1]^{n \times n}) \\ \infty & (\Gamma \notin [0,1]^{n \times n}) \end{cases}. \tag{18}$$

Then, $-\psi(\Gamma)$ is convex and (17) can be solved using a proximal gradient method.

Let $\Gamma_t \in \mathbb{R}^{n \times n}$ denote the solution on the $t$-th step. Let $\mathbf{x}_t$ be the optimal solution of (15) with $\Gamma = \Gamma_t$, i.e.,

$$\mathbf{x}_t \in \arg\min_{\mathbf{x} \in R} \{\mathbf{x}^\top A^- \mathbf{x} + \mathbf{b}^\top \mathbf{x} + h_{\Gamma_t}(\mathbf{x})\}. \tag{19}$$

The partial derivative of $-h_\Gamma(\mathbf{x})$ w.r.t. $\Gamma$ at $(\Gamma_t, \mathbf{x}_t)$, denoted by $S_t$, is then a subgradient of $-\psi(\Gamma)$ at $\Gamma_t$, which can be computed as follows:

$$S_t = A^+ \circ (\mathbf{1}\mathbf{1}^\top - \mathbf{x}_t \mathbf{1}^\top - \mathbf{1}\mathbf{x}_t^\top) \tag{20}$$

**Algorithm 2** An iterative relaxation algorithm for (4)

---

**Input:** Problem instance $(A, \mathbf{b}, C)$ of (4).
**Output:** An approximate solution $\hat{\mathbf{x}}$ to (4) satisfying (25), a lower bound $\psi$ of optimal value of (4).
 1: Set $\Gamma_1 = \mathbf{1}\mathbf{1}^\top/2$, $t = 1$, min_value $= \infty$, $\psi = -\infty$.
 2: **while** Not converged **do**
 3:      Compute $\mathbf{x}_t$ satisfying (19) by using Algorithm 1, and compute

$$\text{value}_t = \mathbf{x}_t^\top A \mathbf{x}_t + \mathbf{b}^\top \mathbf{x}_t, \quad \psi_t = \mathbf{x}_t^\top A^- \mathbf{x}_t + \mathbf{b}^\top \mathbf{x}_t + h_{\Gamma_t}(\mathbf{x}_t), \quad \psi = \max\{\psi, \psi_t\}$$

 4:      **if** value$_t$ < max_value **then**
 5:          Update value and $\hat{\mathbf{x}}$ by

$$\text{min\_value} = \text{value}_t, \quad \hat{\mathbf{x}} = \mathbf{x}_t. \tag{24}$$

 6:      **end if**
 7:      Compute $\Gamma_{t+1}$ by (22) and (23).
 8: **end while**
 9: Return $\hat{\mathbf{x}}$, min_value and $\psi$.

---

By using $S_t$ and a decreasing sequence $\{\eta_t\}$ of positive real numbers, we can express the update scheme for the proximal gradient method as follows:

$$\Gamma_{t+1} \in \arg\min_{\Gamma \in \mathbb{R}^{n \times n}} \left\{ S_t \cdot \Gamma + \frac{1}{2\eta_t} \|\Gamma - \Gamma_t\|^2 + \Omega(\Gamma) \right\}, \tag{21}$$

We can compute $\Gamma_{t+1}$ satisfying (21) by

$$\Gamma_{t+1} = \text{Proj}_{[0,1]^{n \times n}}(\Gamma_t - \eta_t S_t), \tag{22}$$

where $\text{Proj}_{[0,1]}(X)$ is defined by

$$(\text{Proj}_{[0,1]}(X))_{uv} = \begin{cases} 0 & ((X)_{uv} < 0) \\ 1 & ((X)_{uv} > 1) \\ (X)_{uv} & (\text{otherwise}) \end{cases}. \tag{23}$$

The proposed algorithm can be summarized as Algorithm 2.

The choice of $\{\eta_t\}$ has a major impact on the rate of the convergence of the algorithm. From a convergence analysis of the proximal gradient method, when we set $\eta_t = \Theta(1/\sqrt{t})$, it is guaranteed that $\psi_t$ converge to the optimal value $\psi_*$ of (16) and $|\psi_t - \psi_*| = O(1/\sqrt{t})$. Because $\psi(\Gamma)$ is non-smooth and not strongly concave, there is no better guarantee of convergence rate, to the best of our knowledge. In practice, however, we can observe the convergence in $\sim 10$ steps iteration.

### 4.5 Initialization of $\Gamma$

Let $\tilde{\mathbf{x}}_\Gamma$ denote an optimal solution to (15). We employ $\Gamma_1 = 1/2\mathbf{1}\mathbf{1}^\top$ for the initialization of $\Gamma$ because $(x_u + x_u - 1)/2$ is the tightest lower bound of $x_u x_v$ in the max-norm sense, i.e., $h(x_u, x_v) = (x_u + x_v - 1)/2$ is the unique minimizer of $\max_{x_u, x_v \in \{0,1\}} \{|x_u x_v - h(x_u, x_v)|\}$, subject to the constraints that $h(x_u, x_v)$ is affine and bounded from above by $x_u x_v$. In this case, $\tilde{\mathbf{x}}_\Gamma$ is an approximate solution satisfying the following performance guarantee.

**Proposition 3.** *If* $\Gamma = \mathbf{1}\mathbf{1}^\top/2$, *then* $\tilde{\mathbf{x}}_\Gamma$ *satisfies*

$$\tilde{\mathbf{x}}_\Gamma^\top A \tilde{\mathbf{x}}_\Gamma + \mathbf{b}^\top \tilde{\mathbf{x}}_\Gamma \leq \mathbf{x}_*^\top A \mathbf{x}_* + \mathbf{b}^\top \mathbf{x}_* + \frac{1}{2}\mathbf{1}^\top A^+ \mathbf{1}, \tag{25}$$

*where* $\mathbf{x}_*$ *is an optimal solution to* (4).

## 5 Experiments

### 5.1 Simulations

This section investigates behavior of Algorithm 2 on the basis of the simulation model used in [9], and we compare the proposed method with state-of-the-art methods: the SDP relaxation method [9]

Table 1: Ranges of parameters in regression models. (i) is supermodular, (ii) is supermodular + submodular, and (iii) is submodular.

|  | $\beta_{ij}\ (i \neq j)$ | $\beta_{ii}$ | $\alpha_i$ |
|---|---|---|---|
| (i) | $[0, 2]$ | $[-2M, -M]$ | $[M, 3M]$ |
| (ii) | $[-25, 25]$ | $[-2M, 0]$ | $[M, 3M]$ |
| (iii) | $[-2, 0]$ | $[M - 3, M - 1]$ | $[1, 3]$ |

Table 2: Results on real retail data. (a) is computational time, (b) is estimated gross profit, (c) is upper bound.

|  | actual | proposed | QPBO |
|---|---|---|---|
| (a) | - | 36[s] | 964[s] |
| (b) | 1403700 | 1883252 | 1245568 |
| (c) | - | 1897393 | 1894555 |

and the QPBO and QBPOI methods [11]. We use SDPA 7.3.8 to solve SDP problems[7] and use the implementation of QPBO and QPBOI written by Kolmogolov.[8] QPBO methods computes partial labeling, i.e., there might remain unlabeled variables, and we set unlabeled variables to 0 in our experiments. For computing a minimum $s$-$t$ cut, we use Dinic's algorithm [6]. All experiments were conducted in a machine equipped with Intel(R) Xeon(R) CPU E5-2699 v3 @ 2.30GHz, 768GB RAM. We limited all processes to a single CPU core.

**Revenue simulation model [9]** The sales quantity $q_i$ of the $i$-th product was generated from the regression model $q_i = \alpha_i + \sum_{j=1}^{M} \beta_{ij} p_j$, where $\{\alpha_i\}$ and $\{\beta_{ij}\}$ were generated by uniform distributions. We considered three types of uniform distributions to investigate the effect of submodularity, as shown in Table 1, which correspond to three different situations: (i) all pairs of products are substitute goods, i.e., the gross profit function is supermodular, (ii) half pairs are substitute goods and the others are complementary goods, i.e., the gross profit function contains submodular terms and supermodular terms, and (iii) all pairs are complementary goods, i.e., the gross profit function is submodular. Price candidates $\mathcal{P}_i$ and cost $c_i$ for each product are fixed to $\mathcal{P}_i = \{0.6, 0.7, \ldots, 1.0\}$ and $c_i = 0$, respectively.

**Scalability and accuracy comparison** We evaluated four methods in terms of computational time (sec) and optimization accuracy (i.e. optimal values calculated by four methods). In addition to calculating approximate optimal solutions and values, all four algorithms derive upper bounds of *exact* optimal value, which provide information about how accurate the calculated solution.[9] Fig. 1 shows the results with $M = 30, 60, \ldots, 300$ for situations (i),(ii) and (iii). The plotted values are arithmetic means of 5 random problem instances. We can observe that proposed, QPBO and QPBOI methods derived exact solutions in the case (i), which can be confirmed from the computed upper bounds coinciding with the values of objective function. For situations (ii) and (iii), on the other hand, the upper bound and the objective value did not coincide and the solutions by QPBO were worse than the others. The solutions by QPBOI and SDPrelax are as good as the proposed methods, but their computational costs are significantly higher especially for the situations (ii) and (iii). For all situations, the proposed method successfully derived solutions as good as the best of the four methods did, and its computational cost was the lowest.

## 5.2 Real-world retail data

**Data and settings** We applied the proposed method to actual retail data from a middle-size supermarket located in Tokyo [23].[10] We selected 50 regularly-sold beer products. The data range is approximately three years from 2012/01 to 2014/12, and we used the first 35 months (1065 samples) for training regression models and simulated the best price strategy for the next 20 days. Therefore, the problem here was to determine 1000 prices (50 products × 20 days).

For forecasting the sales quantity $q_i^{(d)}$ of the $i$-product on the $d$-th day, we use prices features $\{p_j^{(d')}\}_{1 \leq j \leq 50, d-19 \leq d' \leq d}$ of 50 products for the 20 days before the $d$-th day. In addition to these 1000 linear price features, we employed "day of the week" and "month" features (both binary), as well as temperature forecasting features (continuous), as external features. The price candidates

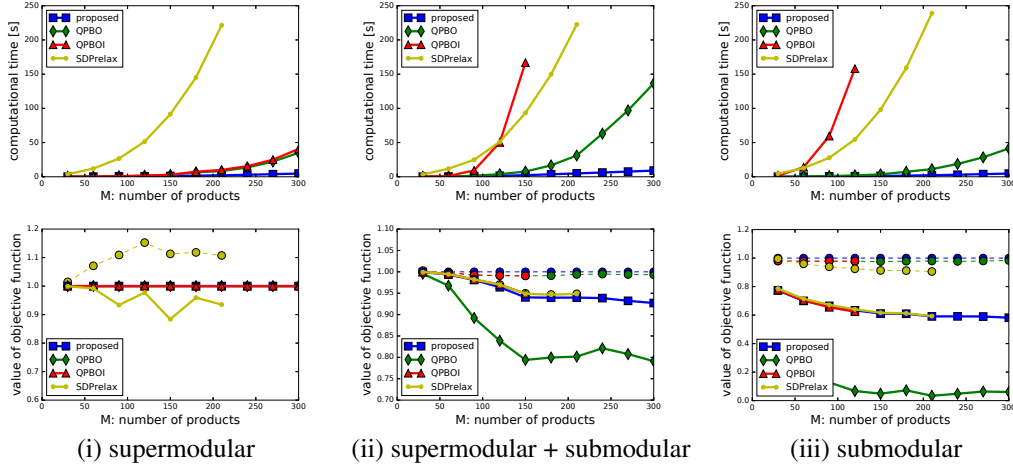

| (i) supermodular | (ii) supermodular + submodular | (iii) submodular |

Figure 1: Comparisons of proposed, QPBO, QPBOI, and SDPrelax methods on revenue simulation data. The horizontal axis represents the number $M$ of products. The vertical axes represent computational time (top) and optimal values of four methods (3) (bottom). For the bottom, circle markers with dashed line represent the computed upper bounds of the optimal values, and optimal values and upper bounds are normalized so that upper bounds with the proposed method are equal to $1$.

$\{P_{ik}^{(d)}\}_{k=1}^{5}$ were generated by splitting equally the range $[P_{i1}, P_{i5}]$, where $P_{i1}$ and $P_{i5}$ are the highest and lowest prices of the $i$-th product in the historical data. We assumed that the cost $c_i^{(d)}$ was $0.3P_{i5}$ (30% of the list prices). Our objective was to obtain a price strategy for 50-products over the 20 days, from the 1066-th to 1085-th, which involves 1000-dimensional variables, in order to maximize the sum of the gross profit for the 20 days. We estimated parameters in regression models, using the ridge regression method. The estimated model contained 310293 pairs with the substitute-goods property and 189207 pairs with complementary goods property.

The results are summarized in Table 2, where "actual" means the gross profit computed on the basis of the historical data regarding sales quantities and prices over the 20 days, from the 1066-th to 1085-th, and costs $c_i^{(d)} = 0.3P_{i5}$. Thus, the target is to find a strategy that expectedly achieves better gross profit than "actual". We have omitted results for QPBOI and SDPrelax here because they did not terminate after running over 8 hours. We observe that the proposed method successfully derived a price strategy over 1000 products, which can be expected to increase gross profit significantly in spite of its cheap computational cost, in contrast to QPBO, which failed with more expensive computation. Although Table 2 shows results using a single CPU core for fair comparison, the algorithm can be easily parallelized that can finish optimization in a few seconds. This makes it possible to dynamically change prices in real time or enables price managers to flexibly explore a better price strategy (changing a price range, target products, domain constraints, etc.)

## 6 Conclusion

In this paper we dealt with price optimization based on large-scale demand forecasting models. We have shown that the gross profit function is supermodular under the assumption of the substitute-goods property. On the basis of this supermodularity, we have proposed an efficient algorithm that employs network flow algorithms and that returns exact solutions for problems with the substitute-goods property. Even in case in which the property does not hold, the proposed algorithm can efficiently find approximate solutions. Our empirical results have shown that the proposed algorithm can handle hundreds/thousands products with much cheaper computational cost than other existing methods.

## Footnotes

[1]"Complementary goods" and "substitute goods" are terms in economics. A good example of complementary goods might be wine and cheese, i.e., if we discount wine, the sales of cheese will increase. An example of substitute goods might be products of different brands in the same product category. If we discount one product, sales of the other products will decrease.

[2]We say that a function $f : \mathcal{D}_1 \times \cdots \times \mathcal{D}_n \to \mathbb{R}$ $(\mathcal{D}_j \subseteq \mathbb{R})$ is *submodular* if $f(\mathbf{x}) + f(\mathbf{y}) \leq f(\mathbf{x} \vee \mathbf{y}) + f(\mathbf{x} \wedge \mathbf{y})$ for all $\mathbf{x}, \mathbf{y}$, where $\mathbf{x} \vee \mathbf{y}$ and $\mathbf{x} \wedge \mathbf{y}$ denote the coordinate-wise maximum and minimum, respectively. We say a function $f$ is *supermodular* if $-f$ is submodular.

[3]The computational cost of the minimum cut depends on the choice of algorithms. For example, if we use Dinic's method, the time complexity is $O(n^3 \log n) = O((KM)^3 \log(KM))$.

[4] $s, t$ are auxiliary vertices different from $1, \ldots, n$ corresponding to source, sink in maximum flow problems.

[5] For details about the construction of $G$, see, e.g., [4, 12].

[6]Problem (16) can be also solved using the ellipsoid method, which guarantees polynomial time-complexity in the input size. However, it is known that the order of its polynomial is large and that the performance of the algorithm can be poor in practice, especially for large size problems. To try to achieve more practical performance, this paper proposes a proximal gradient algorithm.

[7] http://sdpa.sourceforge.net/

[8] http://pub.ist.ac.at/~vnk/software.html

[9] For example, the coincidence of the upper bound and the calculated optimal value implies that the algorithm computed the exact optimal solution.

[10] The Data has been provided by KSP-SP Co., LTD, http://www.ksp-sp.com.

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
