[Supplementary Material]

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

# Appendix

(proof of Proposition 2)

*Proof.* Since the sum of supermodular function is supermodular[8], it suffices to show that $\ell_{ij}$ and $\ell_i$ are supermodular for all $i$ and $j$, under the assumption of the substitute goods property. First, $\ell_i$ are supermodular because arbitrary univariate functions are supermoludar by the definition. Next, let us show the supermodularity of $\ell_{ij}$. Let $p_i, p'_i$ and $p_j, p'_j$ be arbitrary elements of $\mathcal{P}_i$ and $\mathcal{P}_j$, respectively. Denote $\underline{p}_i = \min\{p_i, p'_i\}$, $\overline{p}_i = \max\{p_i, p'_i\}$, $\underline{p}_j = \min\{p_j, p'_j\}$, and $\overline{p}_i = \max\{p_j, p'_j\}$. Without loss of generality, assume $p_i \leq p'_i$. If $p_j \leq p'_j$, then we have $\ell_{ij}(p_i, p_j) + \ell_{ij}(p'_i, p'_j) - \ell_{ij}(\overline{p}_i, \overline{p}_j) - \ell_{ij}(\underline{p}_i, \underline{p}_j) = 0$. Otherwise, i.e, if $p_j > p'_j$, then we have

$$\ell_{ij}(p_i, p_j) + \ell_{ij}(p'_i, p'_j) - \ell_{ij}(\overline{p}_i, \overline{p}_j) - \ell_{ij}(\underline{p}_i, \underline{p}_j)$$
$$= \ell_{ij}(p_i, p_j) + \ell_{ij}(p'_i, p'_j) - \ell_{ij}(p'_i, p_j) - \ell_{ij}(p_i, p'_j)$$
$$= (p_i - p'_i)(f_{ij}(p_j) - f_{ij}(p'_j)) \leq 0,$$

where the last inequality comes from the substitute goods property. These inequality implies the supermodularity of $\ell_{ij}$. $\qquad\square$

(proof of Proposition 3)

*Proof.* From (14), we have

$$\mathbf{x}_*^\top A \mathbf{x}_* + \mathbf{b}^\top \mathbf{x}_* \geq \mathbf{x}_*^\top A^- \mathbf{x}_* + \mathbf{b}^\top \mathbf{x}_* + h_\Gamma(\mathbf{x}_*). \tag{26}$$

Further, it holds that

$$\mathbf{x}_*^\top A^- \mathbf{x}_* + \mathbf{b}^\top \mathbf{x}_* + h_\Gamma(\mathbf{x}_*) \geq \tilde{\mathbf{x}}_\Gamma^\top A^- \tilde{\mathbf{x}}_\Gamma + \mathbf{b}^\top \tilde{\mathbf{x}}_\Gamma + h_\Gamma(\tilde{\mathbf{x}}_\Gamma) \tag{27}$$

since $\tilde{\mathbf{x}}_\Gamma$ is an optimal solution to (15). From the definition (14) of $h_\Gamma$, $h_\Gamma(\tilde{\mathbf{x}}_\Gamma)$ with $\Gamma = \mathbf{1}\mathbf{1}^\top/2$ satisfies

$$h_\Gamma(\tilde{\mathbf{x}}_\Gamma) = \tilde{\mathbf{x}}^\top (A^+ \circ \Gamma)\mathbf{1} + \mathbf{1}^\top(A^+ \circ \Gamma)\tilde{\mathbf{x}} - \mathbf{1}^\top(A^+ \circ \Gamma)\mathbf{1}$$
$$= \frac{1}{2}\tilde{\mathbf{x}}^\top A^+ \mathbf{1} + \frac{1}{2}\mathbf{1}^\top A^+ \tilde{\mathbf{x}} - \frac{1}{2}\mathbf{1}^\top A^+ \mathbf{1}$$
$$= \tilde{\mathbf{x}}_\Gamma^\top A^+ \tilde{\mathbf{x}}_\Gamma - \frac{1}{2}\tilde{\mathbf{x}}_\Gamma^\top A^+ \tilde{\mathbf{x}}_\Gamma - \frac{1}{2}(\mathbf{1} - \tilde{\mathbf{x}}_\Gamma)^\top A^+ (\mathbf{1} - \tilde{\mathbf{x}}_\Gamma)$$
$$\geq \tilde{\mathbf{x}}_\Gamma^\top A^+ \tilde{\mathbf{x}}_\Gamma - \frac{1}{2}\mathbf{1}^\top A^+ \mathbf{1}. \tag{28}$$

From (26), (27) and (28), we obtain

$$\mathbf{x}_*^\top A^- \mathbf{x}_* + \mathbf{b}^\top \mathbf{x}_* + h_\Gamma(\mathbf{x}_*) \geq \tilde{\mathbf{x}}_\Gamma^\top A^- \tilde{\mathbf{x}}_\Gamma + \mathbf{b}^\top \tilde{\mathbf{x}}_\Gamma + \tilde{\mathbf{x}}_\Gamma^\top A^+ \tilde{\mathbf{x}}_\Gamma - \frac{1}{2}\mathbf{1}^\top A^+ \mathbf{1}. \tag{29}$$

$$= \tilde{\mathbf{x}}_\Gamma^\top A \tilde{\mathbf{x}}_\Gamma + \mathbf{b}^\top \tilde{\mathbf{x}}_\Gamma - \frac{1}{2}\mathbf{1}^\top A^+ \mathbf{1}. \tag{30}$$

$\qquad\square$