[Reviews · NeurIPS 2016]

Reviewer 1

Summary

This paper considers the problem of maximizing the seller's revenue by finding the optimal pricing strategy, given certain demand models. In particular, a binary quadratic program, first proposed by Ito and Fujimaki, is applied to formulate this price optimization problem. The authors show that the gross profit function (mapping the prices of all items to the revenue) is supermodular if the demand model satisfies the substitute-goods property. In this case the objective function of the binary quadratic programming problem is submodular, which can be reduced to a minimum s-t cut (or max-flow) problem, which is known to be solvable (e.g. Dinic’s algorithm as it is used by the authors in their experiments). In the case that the substitute-goods property fails to hold, the authors provide an approximation algorithm by constructing an approximation function which bounds the objective function from below. Specifically, the objective function of the binary quadratic program is decomposed into a supermodular and submodular terms, and the supermodular part is replaced by another function which is a lower bound. Then, the authors provide a proximal gradient method to compute this lower bound, with the number of iterations bounded by O(1/sqrt(epsilon)). The authors claim that in practice the convergence can be observed in about 10 iterations. Finally, the authors provide many experiments results about their proximal gradient method based algorithm. They show that their algorithm is considerably faster than the previous algorithms, and produces better pricing strategy (Table 2). In addition, they show that (in Figure 1) the approximated value of the objective function is close to the exact optimal value (by deriving upper bound of it).

Qualitative Assessment

For the first part with substitute goods property, unfortunately, I do not see much novelty in it. The binary quadratic programming formulation was already proposed in Ito and Fujimak’s work, while the reduction from the unconstrained binary quadratic programming problem to the minimum cut is “well known” (as also remarked by the authors). For the second part, the idea of providing a lower bound on the objective function and using proximal gradient method is interesting, but its theoretical part is not well-developed. In particular, there is no theoretical analysis regarding “how well” does the lower bound approximate the exact optimal value. Although empirical results given in Figure 1 address this issue, but I was expecting at least an asymptotic bound to be given. Nevertheless, the empirical results given in the last section seems to be convincing. ---- Post discussion comments ---- I have read all the reviews and the corresponding authors’ responses. My sense is that the paper is perhaps somewhat stronger than my initial assessment, even if it is only a modest contribution on the theory side. For the first part of the paper about the reduction to min-cut, after authors’ feedback 1) and 2), I can now see some novelties. I initially had the impression that they are simply applying existing techniques (the last reviewer has the same impression and raises the same issue). I think the authors should focus more on Proposition 2 which reveals the relationship between supermodularity of gross profit function and the substitute-goods property, because I believe this contribution is even more significant. (I am actually convinced that Substitute-Goods Property is important and practical.) The second part of the paper should be the main part, and I think that the theory part could be better developed, and some other reviewers have a similar opinion. The authors claimed that the approximation gap is bounded by (1^T A^+ 1) / 2 in their responses, but I think this is a weak bound, at least in a theoretical sense. The entries of matrix A^+ are related to functions f_{ij} defined in Equation (1), which can be very large, and I think it is not surprising that this piece was omitted. I think this is a reasonable paper. I would say it is definitely borderline, but it would be OK to accept.

Confidence in this Review

2-Confident (read it all; understood it all reasonably well)


Reviewer 2

Summary

An efficient algorithm is proposed for price optimization.

Qualitative Assessment

1. Please justify the model (1) via references. This model form makes certain restrictions on the function q(p,r); how restrictive is this? How much statistical accuracy is lost? 2. I do not see Table 2 ? 3. Typo in abstract: "Even in case in which the assumption does not hold, the proposed algorithm can efficiently find approximate solutions as good as can other state-of-the-art methods, as empirical results show." 4. typo on line 159

Confidence in this Review

2-Confident (read it all; understood it all reasonably well)


Reviewer 3

Summary

This paper considers the problem of finding optimal prices for goods, with the goal of maximizing the total profit. The number of goods purchased is modeled as a demand forcasting curve - the quantity depends on this good's price, prices of other goods and environmental factors. The authors provide a min cut based algorithm when the goods are substitutes. They extend it to inputs where the goods are a mix of substitutes and complements and provide an algorithm that computes the approximately optimal solution.

Qualitative Assessment

The result for the supermodular case based on cut formulation is not surprising. The extension to mixture of supermodular and submodular is really nice. Experimental results show that this algorithm improves on previous algorithms both in terms of running time and accuracy. Language notes: The abstract is a bit verbose, could be edited to make more terse. In the abstract line 4: rephrase "a number of multiple products" -> "a large number of products" (number and multiple are some what synonymous) Line 18: "much literature" -> "a lot of literature" Line 19: "such industries as" -> "industries such as" Line 23: "such multl-product relationships as " -> "multi-product relationships such as" Line 54: "can" -> "" Line 60: "much literature " -> "a lot of literature" Line i

Confidence in this Review

2-Confident (read it all; understood it all reasonably well)


Reviewer 4

Summary

This paper studied the price optimization based on large-scale demand forecasting models. Under the substitute-goods assumption, the authors have shown the gross profit function is supermodular, and they proposed an efficient and accurate (no approximation) algorithm based on network flow algorithms.

Qualitative Assessment

Price optimization is an important problem, and it usually involves submodular optimization which is NP-hard. There exists approximation algorithm with polynomial time complexity, but the cost in practice is very large when the scale becomes large. The authors reduced the combinatorial optimization problem (3) to binary quadratic programming problem (4). Then they convert the problem parameters to a positive weighted directed graph, after which the exact solution to the binary quadratic problem is solved by finding the minimum s-t cut from the graph when the substitute-good assumption holds. When the assumption does not hold, they proposed a relaxation algorithm which approximates the optimal solution with certain performance guarantee. Strengths: 1. Solved an existing problem from a different view. Used a novel methodology (with connection to network flow algorithms) and achieved a faster algorithm with more accurate results (compared to existing ones -- QPBO, QPBOI, SDPrelax). 2. Problems and the solutions are presented with a clear logic flow. The presentation was precise and clear. 3. The simulation does show less computation time and better accuracy. Weaknesses: 1. It seems the asymptotic performance analysis (i.e., big-oh notation of the complexity) is missing. How is it improved from O(M^6)? 2. On line 205, it should be Fig. 1 instead of Fig. 5.1. In latex, please put '\label' after the '\caption', and the bug will be solved.

Confidence in this Review

2-Confident (read it all; understood it all reasonably well)


Reviewer 5

Summary

This paper provide faster algorithms for price optimization problems. The previous results on this has a large running time of O(M^6). This paper solve this problem using min cut (max flow), and thus improves the running time by a lot, while it does not decreases the performance. It seems that they are directly applying known techniques that translates binary quadratic programs to minimum cut problem. I ask the authors if this is correct or they have significant contribution on translating a new binary quadratic program to minimum cut? This paper have a poor quality of presentation. I expect to see a high level and specific definition of the problem early in the paper. But I could not find it in the first two section of the paper (a page and a half). There is no 'definition' or 'our results' section, so it doesn't get clear what is their problem or contribution early in the paper.

Qualitative Assessment

How do you compare your paper with the previously known techniques (especially reference [12] of your paper)? Are you simply applying their technique, or you have some significant novelty?

Confidence in this Review

1-Less confident (might not have understood significant parts)


Reviewer 6

Summary

The paper develops an efficient algorithm for large scale multi-product price optimization problems that can handle hundreds of products as well as flexible demand models. The paper reveals the connection between submodularity of the revenue and cross elasticity of demand, and proposes a practical, efficient algorithm that employs network flow algorithms for minimum cut problems and returns exact solutions for problems with the substitute-goods property.

Qualitative Assessment

The paper develops an efficient algorithm for large scale multi-product price optimization problems that can handle hundreds of products as well as flexible demand models. One contribution of the paper is to reveal the connection between submodularity of the revenue and cross elasticity of demand. The authors should explain why Substitute-Goods Property is important or practical. When Substitute-Goods Property does not hold, empirical results show that the proposed algorithm can successfully handle hundreds of products and derive solutions as good as can other state-of-the-art methods, with lower computational cost. But there is no analysis for this more general case. To me, theoretical contribution of the paper seems not enough.

Confidence in this Review

1-Less confident (might not have understood significant parts)